# Atomic-resolution imaging of electrically induced oxygen vacancy migration and phase transformation in SrCoO$_{2.5-\sigma}$

Qinghua Zhang[1,2], Xu He[1], Jinan Shi[1], Nianpeng Lu[3], Haobo Li[3],
Qian Yu[4], Ze Zhang[4], Long-Qing Chen[5], Bill Morris[6], Qiang Xu[7], Pu Yu[3,8,9],
Lin Gu[1,8,10], Kuijuan Jin[1,10] & Ce-Wen Nan[2]

Oxygen ion transport is the key issue in redox processes. Visualizing the process of oxygen ion migration with atomic resolution is highly desirable for designing novel devices such as oxidation catalysts, oxygen permeation membranes, and solid oxide fuel cells. Here we show the process of electrically induced oxygen migration and subsequent reconstructive structural transformation in a SrCoO$_{2.5-\sigma}$ film by scanning transmission electron microscopy. We find that the extraction of oxygen from every second SrO layer occurs gradually under an electrical bias; beyond a critical voltage, the brownmillerite units collapse abruptly and evolve into a periodic nano-twined phase with a high c/a ratio and distorted tetrahedra. Our results show that oxygen vacancy rows are not only natural oxygen diffusion channels, but also preferred sites for the induced oxygen vacancies. These direct experimental results of oxygen migration may provide a common mechanism for the electrically induced structural evolution of oxides.

[1] Beijing National Laboratory for Condensed Matter Physics, Institute of Physics, Chinese Academy of Sciences, Beijing 100190, China. [2] State Key Lab of New Ceramics and Fine Processing, School of Materials Science and Engineering, Tsinghua University, Beijing 100084, China. [3] State Key Laboratory of Low-Dimensional Quantum Physics, Department of Physics, Tsinghua University, Beijing 100084, China. [4] Center of Electron Microscopy and State Key Laboratory of Silicon Materials, School of Materials Science and Engineering, Zhejiang University, Hangzhou 310027, China. [5] Department of Materials Science and Engineering, The Pennsylvania State University, University Park, Pennsylvania 16802, USA. [6] Department of Materials Science and Engineering, UC Berkeley, Berkeley, California 94706, USA. [7] DENSsolutions, Informaticalaan 12, Delft 2628ZD, The Netherlands. [8] Collaborative Innovation Center of Quantum Matter, Beijing 100190, China. [9] RIKEN Center for Emergent Matter Science (CEMS), Saitama 351-198, Japan. [10] School of Physical Sciences, University of Chinese Academy of Sciences, Beijing 100049, China. Qinghua Zhang, Xu He, Jinan Shi, and Nianpeng Lu contributed equally to this work. Correspondence and requests for materials should be addressed to Q.Y. (email: yu_qian@zju.edu.cn) or to L.G. (email: l.gu@iphy.ac.cn) or to C.-W.N. (email: cwnan@tsinghua.edu.cn)

Oxygen stoichiometry plays a ubiquitous role in the structure evolution and property optimization of oxide materials including catalytic activity, oxygen ionic conductivity, oxygen permeability, etc[1-4]. Oxygen vacancies ($V_O$s), as one of the most common ion defects, can be effectively employed to tune the functionality of transition metal oxides[5] such as superconductivity of cuprates[6], induced polarization[7, 8], metal-insulator transitions[9-11] and distinct chemical expansion[12]. Specifically, it is considered that the dynamic behaviour of $V_O$s, especially the electromigration of $V_O$s, determines the performance of oxide-based electroresistive memories. For example, an electric field may alter the concentration of oxygen vacancies (Vos) as well as the distribution of oxygen atoms in a crystal lattice, leading to structural instability such as phase separation and thus changes in crystallographic symmetry and functionalities of transition metal oxides such as resistance switching[13]. However, although several hypotheses have been proposed, experimentally characterizing chemical element migration such as oxygen, and crystal structural evolution under an electric field with atomic resolution has been very challenging.

Benefiting from the enhanced image resolution of aberration-corrected transmission electron microscopy (TEM), now it is feasible to characterize the details of oxygen sublattices such as octahedral distortions and local oxygen occupation, at an atomic scale by angular bright-field (ABF) imaging[14, 15] or negative Cs imaging (NCSI) technique[16], which effectively provides new insights into Vo distribution and the related improved electrical and magnetic properties. However, the atomic-resolution studies so far have been all static. Despite highly desirable, in situ characterization of oxygen, and crystal structural evolution under an electric field with atomic resolution has not been achieved to date mainly due to the technical challenge. Therefore, previous experimental approaches of dynamic oxygen ion migration were based on indirect analysis on the change of lattice or spectrum at the nanometer scale or above[17-19].

Here we construct a stable platform for in situ characterization of oxygen at atomic scale by adopting a combination of chip-biased configuration, focused ion beam (FIB) milling and aberration corrected scanning transmission electron microscopy (STEM). The mechanical instability introduced by movable portions in the conventional in situ holders with an electrical bias was avoided, allowing the in situ atomic resolution imaging. With this approach, we investigated the electric-induced oxygen migration and subsequent phase transformation in a $SrCoO_{2.5-\sigma}$ film.

## Results

**Experimental set-up for in situ electrical bias in a STEM.** We selected brownmillerite-type $SrCoO_{2.5}$ as a model system because of its intrinsic oxygen-vacancy ordering, rich polyhedral configurations, and well-studied chemical expansion[20, 21], which can be described as alternating stacks of $[CoO_4]$ tetrahedral and $[CoO_6]$ octahedral layers, with Sr atoms located in the interstitial sites between polyhedra as shown in Fig. 1a. The setup for the in-situ TEM study is shown in Fig. 1b,c. A cross-section of the 50-nm $SrCoO_{2.5-\sigma}$ film was thinned to electron transparency and fixed on the bias chip for in-situ TEM observations, so that we can simultaneously survey resistance states and record corresponding structure changes when applying an electric field. The $SrCoO_{2.5-\sigma}$ film was pre-treated to the low-resistance state (LRS) using the voltage-current cycle shown in Supplementary Fig. 1. The current–voltage (I–V) curve (Fig. 1d) is nearly linear behavior in the

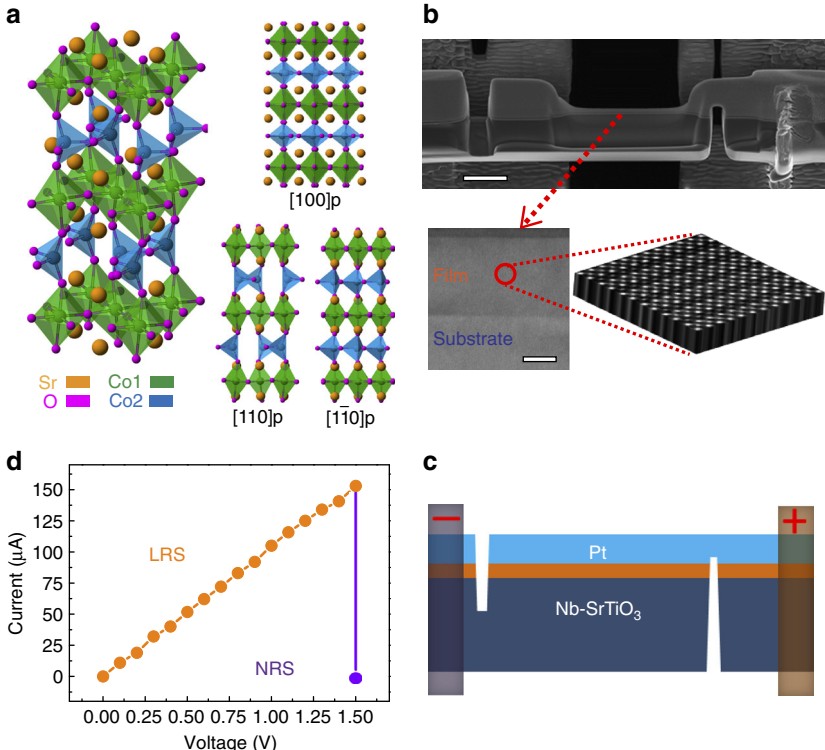

Sr ■ Co1 ■
O ■ Co2 ■

**Fig. 1** Configuration of the experimental set-up. **a** Structure models of $SrCoO_{2.5}$ composed of alternating $[CoO_6]$ octahedra (*green*) and $[CoO_4]$ tetrahedra (*blue*), and their projections along three pseudo-cubic directions [100]p, [110]p and [1–10]p. Scanning electron microscopy (SEM) image (**b**) with a scale bar 2 μm and its schematic (**c**) show the experimental setup of applying an electric field normal to the film in a TEM. The top electrode and film were cut off at the negative side while the bottom electrode (Nb-doped SrTiO₃: Nb-SrTiO₃) and film were cut off at the positive side, producing in a normal electric field on the film. The inserted low-magnification TEM image (scale bar is 20 nm) displays positions of film and substrate. The enlarged stereoscopic HAADF image, as indicated by the *red dotted lines*, demonstrates atomic-resolution *in situ* imaging capability. **d** I–V curve measured in the TEM

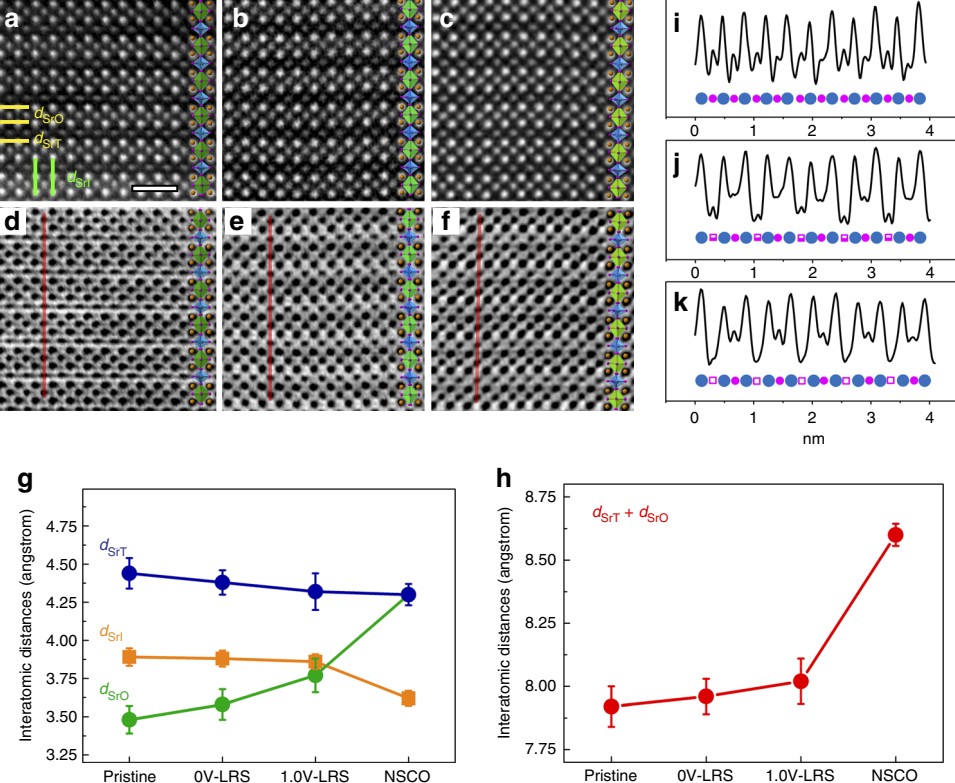

**Fig. 2** Evolution of lattice and oxygen occupancy in three states. HAADF and ABF images for the pristine state (**a**, **d**), LRS (**b**, **e**) and 1V-LRS (**c**, **f**), and related analysis on Sr–Sr distances (**g**, **h**) and image intensities (**i**, **j**, **k**). The contrast correspondence is demonstrated by the structural model overlapping in each image (**a–f**). Two types of out-of-plane Sr–Sr distances: $d_{SrO}$ and $d_{SrT}$, are indicated by three horizontal *yellow lines*. The in-plane Sr–Sr distance $d_{SrI}$ is marked by two vertical *green lines*. **g** Statistics of the interatomic distances of four states, NSCO denotes newly formed $SrCoO_{2.5-\sigma}$ phase. **h** Statistics on the sum of $d_{SrT}$ and $d_{SrO}$. The error bars are the standard error calculated in OriginPro, obtained from five HAADF images in each phase. Line profiles of inversed ABF contrast (**i**, **j** and **k**) corresponding to *red lines* in ABF images (**d**, **e** and **f**), respectively. Small *pink spheres*, half-filled and nearly hollow squares are used to indicate the oxygen occupation. The Co columns are shown as big *blue spheres*. The scale bar in **a** is 1 nm

LRS. When the voltage reaches 1.5 V, the current decreases suddenly from 153 to 1.25 μA, indicating a possible change in $V_O$s under this electric field.

**Atomic-resolution STEM study of $V_O$ migration.** We performed high-angle annular dark-field (HAADF) and angular bright-field (ABF) imaging of $SrCoO_{2.5-\sigma}$ films in the pristine state, the LRS and the LRS under 1 V as shown in Fig. 2, to estimate the oxygen occupation in the $CoO_x$ and $SrO_y$ layers before the phase transformation. In the oxygen-deficient perovskite $ACoO_{3-x}$ (A represents alkaline earth or rare earth elements), a local oxygen deficiency would expand the lattice[20]; a nearly linear relation between the A–A cationic distance and the oxygen deficiency of $CoO_x$ layers has been confirmed[12]. Thus we can estimate the oxygen concentration in the $CoO_x$ planes by measuring two distinctive Sr–Sr distances, $d_{SrT}$ and $d_{SrO}$ (indicated by three horizontal *yellow lines* as shown on HAADF images in Fig. 2a–c), which correspond to the spacings of tetrahedral and octahedral layers, respectively. Using the distances $d_{SrT}$ and $d_{SrO}$ of $CoO_2$ and $CoO_1$ in the pristine state as a reference, we estimated the value of x in $CoO_x$ at LRS and 1V-LRS using the linear relationship reported in ref. 12. The results (Supplementary Table 1 and Fig. 2) show that the average value of x in the $CoO_x$ layer decreased from $1.48 \pm 0.09$ in the LRS to $1.41 \pm 0.11$ in the 1V-LRS.

On the other hand, the evolution of oxygen ions in the $SrO_y$ layers can be directly seen in the ABF images (Fig. 2d–f) where the dark dots indicate atomic positions. Line profiles of contrast (Fig. 2i–k), reflect the oxygen occupation in the SrO layers in these three states. For the pristine state (Fig. 2d,i), the oxygen intensity in each SrO layer was similar, indicating that all SrO layers have nearly the same oxygen content. However, in the LRS (Fig. 2e), the oxygen intensity shows an obvious periodicity as marked by the half-filled square in Fig. 2j. This periodicity indicates that oxygen atoms were only extracted from every second SrO layer. When the voltage was increased to 1.0 V, a further reduction of oxygen contrast was observed (Fig. 2f,k). The simulated ABF images (Supplementary Fig. 3 and Supplementary Fig. 4) suggest that in the LRS state roughly half of the oxygen sites are filled in every second SrO layer ($x = 0.55 \pm 0.10$; Supplementary Fig. 4b), while the weak contrast for the 1V-LRS suggests that <0.4 of the sites are filled ($x = 0.35 \pm 0.10$). Hence, combining measurements on the HAADF lattice spacings with the ABF intensity profiles, we estimated that the composition changed from $SrCoO_{2.50}$ in the pristine material to $SrCoO_{2.26\pm0.14}$ in the LRS state to $SrCoO_{2.09\pm0.16}$ in the LRS state under 1 V (Supplementary Table 1).

**Nano-twinned NSCO phase.** Raising the voltage to 1.5 V induced a sudden structural phase change from the brownmillerite structure (Fig. 2a) to the new crystal structure shown in Fig. 3a. This new phase has a periodic (110)p twined structure which we call the New-$SrCoO_{2.5-\sigma}$ (NSCO) structure here. In this structure the two Sr–Sr distances, $d_{SrT}$ and $d_{SrO}$, become equal (Fig. 2g). The NSCO has a large c axis length ($c_n$~ 4.30 Å, where the subscript n denotes NSCO) and a large axial ratio ($c_n/a_n = 4.30/3.62 = 1.18$). ABF images of the NSCO structure (Fig. 3c) show a homogenous contrast of oxygen columns for all SrO and CoO layers

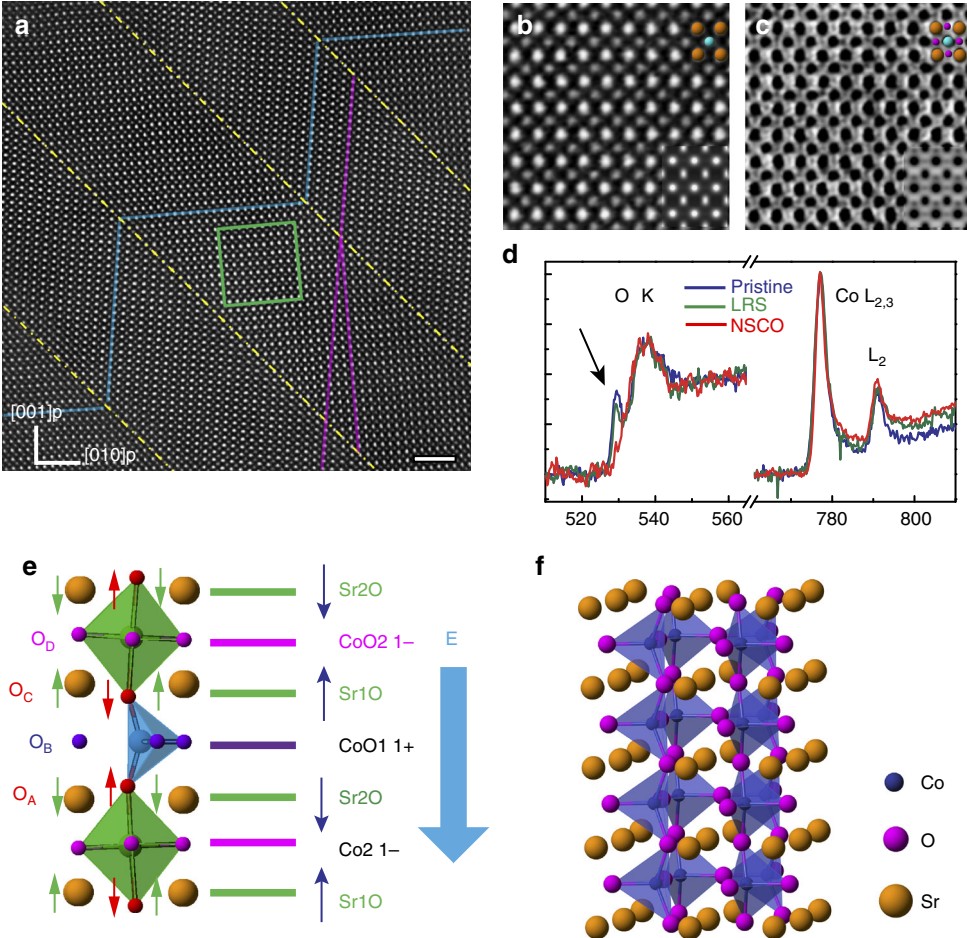

**Fig. 3** Structural information of the NSCO phase and related models. **a** HAADF images of NSCO with periodic nano-twined structures. The twin boundaries are outlined with *yellow dotted lines*. Equivalent layers are marked with *blue lines*. The scale bar is 2 nm. Enlarged HAADF (**b**) and ABF (**c**) images from the green-framed region in **a** after rotation demonstrate. A unit cell is labeled by a set of atomic spheres, where *yellow*, *blue*, and *pink spheres* denote Sr, Co and O atoms, respectively. Simulated HAADF and ABF images were inserted in the corner. **d** Electron energy-loss spectra for three states: O K-edges and Co $L_{2,3}$ edges. A *blue dotted line* highlights the first peak of the O K edges. The peak positions of Co $L_3$ edge in each spectrum have been shifted at same energy position in order to better display the intensity variations of Co $L_3$ edge. **e** Schematic of the atomic electric field and resulting atomic shifts in $SrCoO_{2.5}$. Oxygen ions in the Sr2O, CoO tetrahedral, Sr1O layers, and the CoO octahedral layers are labeled as $O_A$, $O_B$, $O_C$ and $O_D$ respectively. The sheet charge density in each atomic layer was labeled on the right, resulting in an atomic electric field pointing from the octahedral to the tetrahedral layers. *Green* and *red arrows* indicate the atomic displacements of Sr and O in $SrCoO_{2.5}$ from their ideal cubic positions, caused by the atomic electric field. The applied external field E is marked with a big *blue downward arrow*. **f** The calculated structure model of in-phase tetrahedra

(Supplementary Fig. 5), respectively, indicating that the oxygen ions redistributed between the SrO and CoO layers to form the new polyhedral. This oxygen migration and redistribution lead to the elongation along the *c* axis and contraction along the *a* axis that occurs in the transformation to the NSCO structure.

Of interest, a nano-twined structure was formed presumably due to the strain energy relaxation, similar to the formation of twinned domain structure in epitaxial $Pb(Zr_{0.2},Ti_{0.8})O_3$ thin films[22]. The periodicity of the nano-twined structure can be attributed to the substrate clamping effect, where each domain contained an edge dislocation at the interface as shown in Supplementary Fig. 6, which accommodates the lattice mismatch between twinned domains and the Nb-doped $SrTiO_3$ substrate.

Consistent with the atomic structure analysis, the associated electron energy loss spectra also show characteristic changes during the phase transformation in Fig. 3d. The first peak (529.5 eV) at the O K edge decreased when the sample changed from its pristine state to LRS, indicating the creation of $V_O$s[23]. As the structure transformed into the NSCO phase, the first peak at the O K edges disappeared. Valence states of Co in three states:

2.84 ~ 3.08, 2.54 ~ 2.67 and 2.02 ~ 2.15 for the pristine, 0V-LRS and the NSCO phase, were obtained by analysing $L_3/L_2$ intensity ratio of Co with the method reported by Z.L. Wang et al.[24]; details can be found in Supplementary Fig. 7. Here we aligned the peak positions of Co $L_3$ edge in each spectrum to better display the intensity variations of Co $L_2$ edge. In fact, the Co $L_3$ edge shifts towards a lower energy of 774.5 and 772.7 eV in the LRS and NSCO (Supplementary Fig. 8), respectively, indicating an obvious change in the electronic structure of the NSCO phase.

## Discussion

Based on the direct observation of the electro-chemo-mechanically coupled phase transformation presented above, we analyzed the oxygen migration process by finding the possible oxygen extraction sites in pristine $SrCoO_{2.5}$. As shown in Fig. 3e, the atomic electric field pushed the Sr atoms towards the CoO2 layers, but away from the CoO1 layers and created two distinctive Sr–Sr distances, $d_{SrT}$ and $d_{SrO}$. We compared the formation energies of the $V_O$s at different sites. A set of energies (2.57, 2.97

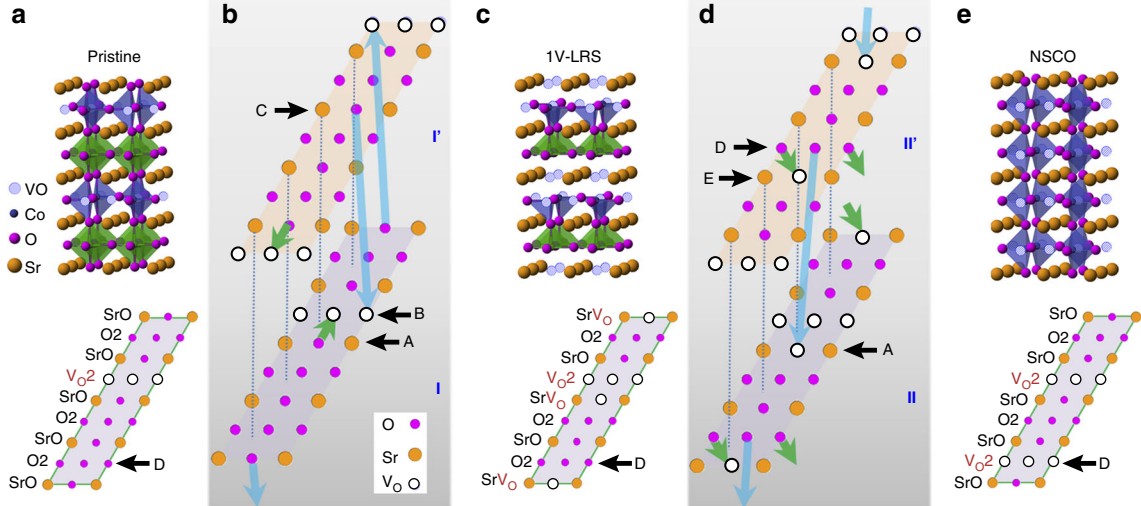

**Fig. 4** Oxygen migration paths with $SrO_{3-x}$ (111) layer models. **a** Structure model and $SrO_{2.5}$ (111) layer of pristine $SrCoO_{2.5}$ with a stacking unit of [SrO-O2-SrO-$V_O$2-SrO-O2-SrO-O2-SrO], ignoring the lattice relaxation near the vacant sites. Schematics of oxygen migration path from the pristine $SrCoO_{2.5}$ to the 1V-LRS (**b**), and from the 1V-LRS to NSCO (**d**), respectively. **I** and **I'** present two neighbored $SrO_2$ (111) layers of the pristine $SrCoO_{2.5}$. *Green arrows* represent the oxygen migration within each (111) layer while *light blue arrows* indicate the oxygen migration between layers. **c**. Structure model and $SrO_2$ (111) layer of 1V-LRS with a stacking unit of [$SrV_O$-O2-SrO-$V_O$-$SrV_O$-O2-SrO-O2-$SrV_O$]. **II** and **II'** present two neighbored $SrO_2$ (111) layers of the 1V-LRS. **e**. Structure model and $SrO_2$ (111) layer of NSCO with a stacking unit of [SrO-O2-SrO-$V_O$2-SrO-$V_O$2-SrO-O2-SrO]. The letters A–E are used to label the SrO, $V_O$2, or O2 rows

and 2.72 eV) was obtained for $O_A$ ($O_C$), $O_B$ and $O_D$, indicating that the oxygen ions at the SrO layers ($O_A$ and $O_C$) have a higher tendency to be extracted. When a positive external field was applied, the $O_A$ in the $Sr_2O$ layer experienced a larger positive field due to the additive impact of the external field and preferred to be extracted, which is in good agreement with the phenomenon that oxygen extraction happens in every second SrO layer of the LRS samples.

With systematic theoretical calculations, we concluded that the in-phase distorted tetrahedron structure (Fig. 3f) is the most likely one for the NSCO phase with the ideal composition of $SrCoO_2$. The in-phase distorted tetrahedron structure demonstrates a good agreement with the experimental observation as shown in the simulated HAADF and ABF images inserted in Fig. 3b,c. It displays a large c lattice constant (4.13 Å) and c/a ratio (4.13/3.58 = 1.14), which is also close to the experimental values. The small difference between calculated and experimental lattice parameters might be caused by the strain field of NSCO restricted by the substrate and twin boundaries.

Regarding the migration paths of oxygen, we adopted a layer model of perovskite $ABO_{3-x}$ where the perovskite structure is described as a cubic close-packed arrangement of $AO_{3-x}$ layers between which one-fourth of the octahedral interstices are filled by B cations[25]. In this framework, the $SrO_3$ (111) plane of $SrCoO_3$ can be represented by a repeated unit of [SrO-O2] (Supplementary Fig. 9) while structures of pristine $SrCoO_{2.5}$, 1V-LRS and NSCO were decomposed as cubic-close-packed stacking of different $SrO_{3-x}$ layers, as shown in Supplementary Fig. 10 and Fig. 4. At first, Vo rows appear in every three O2 rows in pristine $SrCoO_{2.5}$ (Fig. 4a); as the voltage increases from 0 to 1 V, more Vos were created in the SrO planes in 1V-LRS (Fig. 4c). In NSCO, Vos rearranged and new Vo rows formed (Fig. 4e). Dynamically, the Vo migration paths can be interpreted as following: in an external field, O ions can diffuse from their sits on SrO rows (e.g., the row A in Fig. 4b) to the adjacent Vo rows (e.g., the row B) as indicated by *green arrows*, or diffuse from their sits on SrO rows on layer' to the Vo rows on the nearby layer **I** and vice versa (e.g., from C to B) as indicated by the *light blue arrows* by a single diffusive step in Fig. 4b; later the O ions on the Vo rows

B diffuse out of the crystal easily by moving along the vacant row, leading to the 1V-LRS structure. This is in good agreement with the low activation energy barrier height in the calculation of oxygen diffusion pathways[26]. At a critical voltage of 1.5 V, the O ions further diffuse from the O2 rows (e.g., the row D) to the sits on neighboring $SrV_O$ planes (e.g., the row E) or to the closest $SrV_O$ planes on the neighboring layer (e.g., the row A) as indicated by the *light blue arrows* in Fig. 4d. Thus, another Vo row D forms at the position of original O2 row as shown in Fig. 4e, leading to a transformation to the NSCO lattice. The Vo rows in the $SrO_{3-x}$ (111) layer provide a natural channel for oxygen extraction and also act as a preferred state for the induced Vos.

In summary, by using the state of the art in situ STEM characterization with atomic image resolution, we demonstrated the extraction process of the oxygen ions in $SrCoO_{2.5-\sigma}$ from every second SrO layer under an electric field, which eventually leads to a sudden structural phase transformation at a critical voltage of 1.5 V. $V_O$ rows are not only natural oxygen diffusion channels, but also preferred sites for induced $V_O$s. The directly observed experimental pictures of the oxygen migration process at the atomic scale shed light on the atomistic mechanisms of an electric-induced structural phase transformation through electro-chemo-mechanical coupling, which can be utilized for the design of advanced functional oxide materials.

## Methods

**In situ sample preparation**. The TEM samples were prepared by using Focused Ion Beam (FIB) milling. The cross-sectional lamella were thinned down to 100 nm thick at an accelerating voltage of 30 kV with a decreasing current from the maximum 2.5 nA, followed by a fine polish at an accelerating voltage of 2 kV with a small current of 40 pA. Next, we fixed the thin cross-sectional TEM lamella on the two electrodes of the Nano-Chip by Pt deposition in a FIB system. The top electrode and film were cut at the negative side while the bottom electrode and film were cut at the positive side, resulting in an electric field on the film. This configuration is favorable in exerting a large electric field with a small voltage because of the small thickness of films. The in-situ electric field experiments were performed using a heating and bias DH30 holder produced by DENSsolutions.

**STEM characterization**. The atomic structures of the $SrCoO_{2.5-\sigma}$ films was characterized using an ARM—200CF (JEOL, Tokyo, Japan) transmission electron microscope operated at 200 kV and equipped with double spherical aberration (Cs)

correctors. The attainable resolution of the probe defined by the objective pre-field is 78 picometers. ABF and HAADF images were acquired at acceptance angles of 11–22 and 90–250 mrad, respectively. All of the images presented here are Fourier-filtered to minimize the effect of the contrast noise. The filtering does not have any effect on the results of our measurements. In addition, we adjusted deliberately the brightness and contrast in order to better representation of the atomic arrangements, this adjustment does not have any effect on the result. The EELS experiments were carried out with a Gatan spectrometer attached to the TEM in the STEM mode operating at 200 kV. The convergence semi-angle for the electron probe was 22 mrad. The spectrometer was set to an energy dispersion of 0.25 eV/channel. All of the spectra were acquired with a short exposure time, which was as short as 0.05 s, in a line-scan manner. Three hundred spectra were summed to avoid electron beam damage.

**Density functional theory calculations**. We compared the formation energies of the oxygen vacancies ($V_O$s) at different sites. The formation energy is taken as the difference between the sum of the energies of the structure with one $V_O$ and that of an oxygen atom and the energy of the structure without $V_O$: $E_{form} = E(Sr_{32}Co_{32}O_{79}) + E(O) - E(Sr_{32}Co_{32}O_{80})$. A supercell of $2 \times 2 \times 1$ primitive cells (i.e., $2 \times 2 \times 1$ $Sr_8Co_8O_{20}$ of 144 atoms) was used. One oxygen atom is removed from the supercell to create a $V_O$ in the supercell. The structures were relaxed while the hopping of oxygen ions between sites were not considered. The in-plane lattice parameters were still fixed to the substrate (3.905 angstrom, SrTiO₃). The first principles calculations were performed with PBE generalized gradient approximation (GGA) and projector augmented wave (PAW) method as implemented in Vienna Ab initio Simulation Package (VASP). We used a plane-wave basis set with the energy cutoff of 450 eV and a $n_a \times n_b \times n_c$ Γ centered k-points to integrate the Brillouin zone, where $n_a$, $n_b$, and $n_c$ are about $20/a$, $20/b$, and $20/c$ respectively, and $a$, $b$, $c$ are the lattice parameters in Angstrom. The pseudopotentials used are Sr_sv, Co and O as implemented in the VASP package. A DFT + U correction with an effective $U = 3.5$ eV was used to better describe the on-site Coulomb interaction between Co 3d electrons with an effective $U = 3.5$ eV. The structures are fully relaxed until the residual forces are below $10^{-2}$ eV/Å. The magnetic structures are assumed to be G-type antiferromagnetic, as in the pristine SrCoO₂.₅ with the Ima2 structure.

**Data availability**. All data supporting this study and its findings are available within the article, its Supplementary Information and associated files. Any source data deemed relevant are available from the corresponding author upon request.

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

## Acknowledgements

This work was supported by National Program on Key Basic Research Project (2014CB921002, 2015CB921700) and The Strategic Priority Research Program of Chinese Academy of Sciences (Grant No. XDB07030200) and National Natural Science Foundation of China (51522212, 51421002, 51332001 and 11274194). The effort at Penn state is supported by the U.S. Department of Energy, Office of Basic Energy Sciences, Division of Materials Sciences and Engineering under Award DE-FG02-07ER46417 (L.-Q.C.). Q.Y. acknowledges support by the Chinese 1000-Youth-Talent Plan, 111 project under Grant No. B16042, National Natural Science Foundation of China (51671168) and the State Key Program for Basic Research in China under Grant No. 2015CB65930.

## Author contributions

The project was conceived and directed by L.G., Q.Y., P.Y. and C.-W.N.; Q.Z., Q.Y. and L.-Q.C. wrote the manuscript; TEM experiments were performed and analyzed by Q.H.Z. and L.G. under the guidance of C.-W.N., Z.Z. and B.M.; TEM lamellas were fabricated with FIB milling by J.S. and Q.X.; SrCoO₂.₅ films were grown by N.L. and H.L. under the guidance of P.Y.; The first principle calculations were performed by X.H. under the guidance of K.J.

## Additional information

**Competing interests:** The authors declare no competing financial interests.

