## [Peer Review File · Nature Communications]

Reviewers' comments:

Reviewer #1 (Remarks to the Author):

In this work, the dynamic process of electric-induced oxygen migration and subsequent reconstructive structural transformation in a brownmillerite-type SrCoO_{2.5-σ} film were revealed at atomic scale through scanning transmission electron microscopy with aberration corrector. The work demonstrated the dynamic extraction process of the oxygen ions in SrCoO_{2.5-σ} from every second SrO layer under an electric field, which eventually leads to a sudden structural phase transformation at a critical voltage of 1.5 V. These new insights can promote the understanding on the atomistic mechanisms of an electric-induced structural phase transformation as well as serve as design principles for future optimization of advanced oxide materials

In regard to this manuscript, the reviewer raises the following questions:

1. On line 104 of page 6, the authors need to clarify the notation of dSrT and dSrO.
2. For the estimation of oxygen occupancy in the CoO_x and SrO_y layers, the author provide two different methodologies. Please explain the reason in details and more importantly clarify the changes of Sr-Sr distances can not be influenced by the oxygen vacancies in the SrO_y layer.
3. On line 121 and 122 of page 7, the authors claim oxygen occupancy quantification for different bias states. What about the certainty of the quantification result? Or is there any other characterization can support the quantification result?
4. On line 141 of page 8, the author claims "ABF images of the NSCO structure (Fig. 3c) show a homogenous contrast of oxygen columns for all SrO and CoO layers". The author needs to provide line profiles of inversed ABF contrast like Fig. 2(i), (j) and (k) to validate the point.
5. Between line 154 to 158 of page 9, it is hard to see the changes of pre-edge intensity of O K edge for pristine and LRS state. In addition, the authors should explain why the valence of Co decreases more obvious for LRS structure but only has subtle change from LRS to NSCO structure.
6. On line 186 and 187 of page 11, why the larger positive electric field is, the easier oxygen is to be extracted? What about negative electric field? If it has been demonstrated before, please show references.

Overall Opinion:

There are some grammatical errors and typos scattered throughout this manuscript. The present form is not suitable for publication without major corrections, and authors should answer the questions the reviewer brought up above.

Reviewer #2 (Remarks to the Author):

The paper entitled "Atomic-resolution imaging of electrically induced oxygen vacancy migration and phase transformation in brownmillerite cobalt oxides," by Zhang et al. describes a nice in-situ study of SrCoO₃ thin films under applied bias. The reported STEM images and DFT modeling are impressive, but the authors overstate their case about dynamic measurements in STEM.

While the authors claim to have revealed the dynamic processes of oxygen migration, the paper does not actually report such a thing. Since all the measurements are conducted in STEM mode (which takes several seconds to acquire), there is no dynamic information in this data set. It stands to reason that the same information could have been extracted for samples that underwent ex-situ electrical testing and then were prepared for TEM. Alternatively, in-situ heating (similar to the work by Klie and Browning on LaSrFeO₃, Journal of Electron Microscopy, 2002. 51: p. S59-S66) should have shown the same effect that is reported in this submitted manuscript. In fact, one

can argue that the observed formation (or ordering?) of oxygen vacancies is due to heating more so than due to the electrical field.

In addition to these general comments, here are a few specific questions the authors need to address:

1) In line 57, the authors seem to state that NCSI is a STEM based technique. This is clearly false, since negative CS imaging is only applicable for aberration-corrected TEM imaging.

2) The DFT modeling does not appear to reveal any information that was not previously known. Simply calculating the oxygen vacancy formation energies has been done numerous times in the reported literature.

3) In Figure S2b), I would argue that the fit between the experiment and the modeled ABF images is poor. The contrast in the oxygen columns at 0.5, 1.4 and 2.1 nm is not reproduced at all in the models. If this serves as the foundation for the entire paper, then the authors should revise this model completely.

4) How do the authors distinguish the reported new LCO phase from one that simply shows disordered oxygen vacancies? In a recent paper on PrYCaCoO₃ (Gulec, ACS Nano, 2016. 10(1): p. 938-947), the authors report a temperature induced oxygen vacancy ordering/disordering transition which is measured as a change in the sample conductivity. Is it possible that a similar transition is observed here rather than the formation of a new phase?

Reviewer #4 (Remarks to the Author):

In situ observation of oxygen migration at the atomic scale is a really interesting topic that could answer many questions on the characterization of a variety of materials. Such a novel approach could also provide important information on understanding the properties of systems as catalysts, membranes and fuel cells.

The authors claim to detect oxygen migration in brownmillerite-type SrCoO_{2.5-σ} under electric field however I consider that there are some issues that need further explanation before considering the publication of the paper as a nature communication.

1.- Page 6, (line 96 to 108)

The authors claim to determine the oxygen occupancy in CoO_x and SrO_y layers by using the linear relationship reported in ref 12. It is not clear if they had used the linear relationship published in ref12, which covers a 3.5-4.8nm range or they have provided an experimental linear relationship for their own data (it could be included as supporting information). I think that this information needs to be clarified

2.- Page 6 (line 110-117)

By comparing the ABF intensities on the pristine, LRS and 1V-LRS images, the experimental images show a change on the intensity periodicity between the pristine sample and the LRS and 1V-LRS samples. The authors claim that such a change indicates that oxygen atoms were only extracted from every second Sr-O layer on the LRS sample and a further reduction on the oxygen content is observed when applying 1V.

In order to justify such a result, the authors present ABF image simulations as supporting information. I think that this point needs a further explanation and some points need to be considered before arriving to such a conclusion.

* They are comparing the experimental data Figure 2 i-k with the simulated image information (figure S2a, intensity profiles), however the experimental figure is displayed in nm and the simulated figure is displayed in pixels.

* On the simulated data a sample thickness of 39nm is used. Why are they using such a value? Have they performed simulations at different thickness?

* When comparing the pristine sample with the LRS and 1V-LRS samples, on the experimental data they mentioned a change on the intensity periodicity however if we look at the simulated intensity profiles (Fig S2); such a change is not clearly observed from the pristine sample and the different oxygen occupancy images (0.1-0.9), at least, we need to go lower than 0.5 in order to see such a change in periodicity. I think that a graph comparing all the experimental intensity profiles (pristine, LRS and 1-LRS) with the simulated one will help to understand such a critical point on the paper.

Moreover they indicated that the LRS state will correspond with $x=0.5$ occupancy (Figure S2 b). When looking at the Figure S2b, we observe that there is a discrepancy between the experimental and simulated every two layers (between 0.5-0.75nm, between 1.25 and 1.5nm etc...). On the simulated images it is clear that the intensity corresponding to the first oxygen peak is disappearing when moving from the pristine to the 0.1 occupancy. Why are they indicating that LRS sample corresponds to an occupancy of $x=0.5$ and not for example to 0.6 or 0.7? I think that it is important to clarify this part of the paper.

* Do they have any analytical information (as for example EELS quantification) to support the different oxygen content between the pristine and LRS, 1V-LRS samples?

3.- Page 8 Nano-twinned NSCO phase

When applying a 1.5V the authors reported the formation of a new crystal structure.

On Figure 3, they present EELS data from the pristine, LRS and NSCO samples. I understand that such information has also been extracted from the 1V-LRS sample. How much is the oxygen content on the indicated samples? What are the experimental conditions for the EELS acquisition?

On line 153, "The first peak (529.5 eV) at the O K edge decreased when the sample changed from its pristine state to LRS, indicating the creation of VO₂" In ref 23, Gazquez et al, after analysing the L_{2,3} ratio, they conclude that the oxidation state is close to Co⁺² and it remains unchanged, moreover they conclude that the measured spatial modulation in the O-K edge prepeak is the fingerprint for a spin-state superlattice caused by the ordering of high concentration of O vacancies. On the present paper (line 153) the authors claim that the first peak at the O K edge decreased when the sample change from its pristine state to LRS. Such an observation is not clearly observed on Figure 3d. The figure needs to be modified in order to better appreciate such a change on the figure.

Moreover on figure 3d, they indicated that the L₃ peak of Co goes from 772.5eV on the NSCO sample to 774.50 on LRS sample and 777.20eV on pristine sample indicating that the valence decrease of Co⁺³ on NSCO sample. In this context I have to mentioned that there are several papers that shown that a change in valence state of cations introduces a dramatic change in the L_{2,3} white line ratio (see for example Pearson et al. Appl. Phys. Lett. 53, 1405–1407, 1998, Pearson et al. Phys. Rev. B 47, 8471–8478. 1993; Kurata and Colliex Phys. Rev B, 48, 2102-2108) In particular Z.L. Wang et al. Micron 31 (2000) 571–580) 2000, reported the behavior of several cobalt oxides with oxidation states going from +2 to +4 and they also determine the oxidation state of a perovskite-type crystals. On the mentioned papers and at the EELS spectra published on the EELS database <https://eelsdb.eu/> we can see that the L_{2,3} signal of several Co compounds are on the 780-790eV range. I suggest to the authors to analyze the L_{2,3} ratio in order to determine the oxidation state on their samples and to check the energy value of the Co L_{2,3} signal on the

Figure 3d.

When comparing the Co L_{2,3} on the pristine and LRS samples there is a shift in energy from 777.20eV towards 774.50. What does mean? How they can explain such a modification?

4.- I also consider that some modifications could be included in the figures in order to clarify the diagram. In figure 1 c, it will be interesting to indicate at the figure where the sample SrCoO_{2.5} is located at the experimental set-up.

After reviewing the "Atomic-resolution imaging of electrically induced oxygen vacancy migration and phase transformation in brownmillerite cobalt oxides" paper I consider that the major revision above indicated need to be done in order to strengthen their conclusions

Reply to comments of Referee #1

Remarks to the Author. In this work, the dynamic process of electric-induced oxygen migration and subsequent reconstructive structural transformation in a brownmillerite-type $\text{SrCoO}_{2.5-\sigma}$ film were revealed at atomic scale through scanning transmission electron microscopy with aberration corrector. The work demonstrated the dynamic extraction process of the oxygen ions in $\text{SrCoO}_{2.5-\sigma}$ from every second SrO layer under an electric field, which eventually leads to a sudden structural phase transformation at a critical voltage of 1.5 V. **These new insights can promote the understanding on the atomistic mechanisms of an electric-induced structural phase transformation as well as serve as design principles for future optimization of advanced oxide materials.**

Comment 1. On line 104 of page 6, the authors need to clarify the notation of d_{SrT} and d_{SrO} .

Response 1: We thank the referee for this suggestion. We have added explanations for d_{SrT} and d_{SrO} in the revised manuscript (please see page 6). d_{SrT} and d_{SrO} are the spacings of the octahedral and tetrahedral layers, respectively.

Comment 2. For the estimation of oxygen occupancy in the CoO_x and SrO_y layers, the author provide two different methodologies. Please explain the reason in details and more importantly clarify the changes of Sr-Sr distances cannot be influenced by the oxygen vacancies in the SrO_y layer.

Response 2:

a. Reason for measuring out-of-plane Sr-Sr distances d_{SrT} and d_{SrO} .

In oxides, there is a strong connection between the vacancy concentration and the molar volume of a compound, which is known as chemical expansion. For instance, the $(\text{La,Sr})\text{CoO}_{3-x}$ system has well-established structural defect chemistry, where the lattice expansion is proportional to the concentration of oxygen vacancies. Specifically, as demonstrated in the studies of the $(\text{La,Sr})\text{CoO}_{3-x}$ system, the “interplanar spacing is directly related to oxygen content (as shown in Figs. 2a and 3a in Ref.12); thus the chemical expansion can be used as a universal measurement to determine the oxygen stoichiometry in both ordered and disordered oxygen-deficient materials” (Nat. Mater. 11, 888-894 (2012)). This provides a quantitative approach for calculating the amount of oxygen vacancies in the CoO_x layers.

We used pristine $\text{SrCoO}_{2.5}$ (-SrO-CoO1-SrO-CoO2-) as the reference and extracted the linear coefficient. For $x=1$, $d_{\text{SrT}}=4.44 \text{ \AA}$, and for $x=2$, $d_{\text{SrO}}=3.48 \text{ \AA}$. We assumed a linear relationship between the oxygen content (x) and the lattice spacing (d), and we thus obtained:

$$x=5.625-1.0416*d$$

From this, we can obtain the value of x in CoO_x by measuring the interplanar spacings, d_{SrT} and d_{SrO} . The results of the averaged occupations x in the CoO_x

layers for the LRS and 1V-LRS samples are shown in *Supplementary Table S1* and Fig. R0.

Figure R0. The linear relationship between the Sr-Sr distance and the oxygen occupancy x in the CoO_x layers.

b. Reason for using contrast variation in ABF.

As shown in Fig. 2g, the in-plane Sr-Sr distance d_{Sr} did not change during the transition from pristine state to the LRS state while the change in contrast of the oxygen columns in the SrO planes was obvious. Thus, we directly used the contrast variation in ABF to estimate the oxygen occupation y in the SrO planes. Certainly, this method is limited quantitatively because of factors like the uncertainty in the thickness and because simulations do not perfectly replicate real experimental conditions. However, in our in situ measurements, the experimental conditions and sample thickness were identical. Thus, we believe that this method provides a reasonable estimate for the amount of oxygen vacancies in the SrO planes. We also carried out systematic comparisons of the simulated ABF images to obtain the range of oxygen content in the SrO planes, and this is discussed in detailed in Response 3.

c. Clarify that the changes in the Sr-Sr distances cannot be influenced by the oxygen vacancies in the SrO_y layer.

We would like to clarify that we cannot claim that the oxygen vacancies in the SrO_y layer have no influence on the out-of-plane Sr-Sr distances. However, its influence is negligible. As the sample changed from the pristine state to 1V-LRS, there was a relatively large change in the oxygen content of the SrO layer from 1.0 to 0.35 ± 0.16 . If we attributed this change to the change in the Sr-Sr distance, according to the linear relationship between the oxygen content

and the Sr-Sr distances mentioned above, there would be a very large change of $\sim 0.61 \text{ \AA}$ in the Sr-Sr distance in 1V-LRS, but this was not the case. In fact, as shown in Fig. 2g, there was a decrease in d_{SrT} and an increase in d_{SrO} , with an average expansion of $\sim 0.09 \text{ \AA}$.

Comment 3. On line 121 and 122 of page 7, the authors claim oxygen occupancy quantification for different bias states. What about the certainty of the quantification result? Or is there any other characterization can support the quantification result?

Response 3: As described in Response 2 above, we used x for the CoO $_x$ layers and y for the SrO $_y$ layers to represent the oxygen occupancy. The value of x was directly calculated based on the measured Sr-Sr distances. The x values with error bars were quantified in this way and are shown in *Supplementary Table S1*.

The value of y was obtained by comparing the intensities of the oxygen columns in the ABF images with simulations. Because the ABF contrast can be affected by sample thickness, defocus, aberrations, and misalignments, the two never match exactly. In the revised version, we made further statistical comparisons. For example, as shown in Fig. R1, we set the oxygen occupancy to 0.4, 0.5, and 0.6 and compared the ABF images with the simulations. The 0.5 and 0.6 curves matched the decrease of contrast better as indicated by the green and blue arrows, respectively. Thus, an error bar of 0.1 was given. The value of y was estimated as 0.55 ± 0.1 . All of the results with error estimations can be found in the revised *Supplementary Table S1*.

Figure R1. Comparison of ABF images of the 0V-LRS with the simulations on the oxygen occupancy of 0.4, 0.5 and 0.6 in the SrO planes.

The oxygen content in these states can also be determined from the valence state of Co. In the revised version, we analyzed the $L_{2,3}$ ratio using the method reported by Prof. Z.L. Wang et al. [*Micron* 31 (2000) 571–580]. The three valence states of Co are 2.84~3.08, 2.54~2.67, and 2.02~2.15 for the pristine, 0V-LRS, and NSCO phases, respectively. Details of the quantification analysis of the valence state of Co can be found in Fig. R9. We determined the oxygen content from the valence states of Co

based on the electroneutrality principle of SrCoO_z and obtained the oxygen content z as 2.42~2.54 and 2.27~2.34 for the pristine and 0V-LRS states, respectively. This is in good agreement with the results from the HAADF and ABF images, which are 2.50 ± 0.10 and 2.26 ± 0.14 , respectively. We have added this result in the revised manuscript.

Comment 4. On line 141 of page 8, the author claims “ABF images of the NSCO structure (Fig. 3c) show a homogenous contrast of oxygen columns for all SrO and CoO layers”. The author needs to provide line profiles of inversed ABF contrast like Fig. 2(i), (j) and (k) to validate the point.

Response 4: We have provided the line profiles of the inversed ABF contrast on the CoO layers and SrO layers of the NSCO phase, which demonstrates the homogenous contrast of the oxygen columns for the SrO and CoO layers. As shown in Fig. R2, the contrast of the oxygen columns in the CoO layers is basically homogenous.

Figure R2. Line profiles of inversed ABF contrast of SrO and CoO layers, where red lines indicated positions on the ABF images with corresponding numbers.

Comment 5. Between line 154 to 158 of page 9, it is hard to see the changes of pre-edge intensity of O K edge for pristine and LRS state. In addition, the authors should explain why the valence of Co decreases more obvious for LRS structure but only has subtle change from LRS to NSCO structure.

Response 5: To visualize the change of the O K edge among the three states, we have revised Fig. 3d to show the comparison of the pre-peak of the O K edge and the $L_{2,3}$ ratio of the Co L edges. As the sample changes from its pristine state to LRS, a decrease in the pre-peak at the O K edge was clearly observed, and this is shown in Fig. R3. The previous Fig.3d was also provided in the supplementary materials.

Fig. R3. Electron energy-loss spectra for three states: O K-edges and Co L_{2,3}-edges. The black arrow labels the first peak of the O K edges.

Obvious changes in the pre-peak of the O K edge from the pristine state to 0V-LRS and from 0V-LRS to NSCO were observed and are shown in Fig. R3. In the revised version, we analyzed the $L_{2,3}$ ratio using the method reported by Prof. Z.L. Wang et al. [Micron 31 (2000) 571–580]. The three valence state of Co are 2.84~3.08, 2.54~2.67, and 2.02~2.15 for the pristine, 0V-LRS, and the NSCO phases, respectively. Details of the quantification analysis on the valence state of Co can be found in Fig. R9.

Comment 6. On line 186 and 187 of page 11, why the larger positive electric field is, the easier oxygen is to be extracted? What about negative electric field? If it has been demonstrated before, please show references.

Response 6: Positive and negative electric fields can both extract oxygen. Here we only used a positive electric field as an example, and we found that the O_A sites in the Sr_2O layer were easier to extract since a larger electric force was applied on the oxygen atoms. In fact, since the O_A and O_C sites are equal and have the lowest formation energies of the V_{OS} , when a negative electric field is applied the O_C sites are the most preferred for extraction because the negative electric field is parallel to the atomic electric field of the O_C sites.

Reviewer #2 (Remarks to the Author):

Remarks to the Author. The paper entitled "Atomic-resolution imaging of electrically induced oxygen vacancy migration and phase transformation in brownmillerite cobalt oxides," by Zhang et al. describes a nice in-situ study of **SrCoO₃ thin films under applied bias. The reported STEM images and DFT modeling are impressive, but the authors overstate their case about dynamic measurements in STEM.** While the authors claim to have revealed the dynamic processes of oxygen migration, the paper does not actually report such a thing. Since all the measurements are conducted in STEM mode (which takes several seconds to acquire), there is no dynamic information in this data set. It stands to reason that the same information could have been extracted for samples that underwent ex-situ electrical testing and then were prepared for TEM. Alternatively, in-situ heating (similar to the work by Klie and Browning on LaSrFeO₃, Journal of Electron Microscopy, 2002. 51: p. S59-S66) should have shown the same effect that is reported in this submitted manuscript. In fact, one can argue that the observed formation (or ordering?) of oxygen vacancies is due to heating more so than due to the electrical field.

Response: We thank the referee for recognizing the quality of our work. We agree with the referee that STEM imaging is not the real time record. The time resolution is not enough to record the real dynamic process. However, time resolution and image resolution are hard to balance. For studying the oxygen atom, high image resolution and energy resolution are required. Our experimental method can at least provide good image resolution of the changes with atomic resolution under an applied bias. Even though our work is not truly "in situ", it still provides better information than an ex situ study. This is illustrated by three points in particular. 1) We can image the structure under an applied bias, while for an ex situ study, the sample is prepared with a bias off, and the change in the atomic structure may recover. 2) We have a direct correlation between functionality (e.g., the I-V curve) and the local structures on the nanoscale. Thus we obtained important information including the critical voltage for the phase transformation, which is hard to obtain doing ex situ studies. 3) In the in situ study, we examined the same sample, so factors such as image condition and sample thickness were the same and guaranteed a quantitative analysis and comparison. In an ex situ study, at least the sample thickness cannot be the same every time, and this influences the quantitative analysis. But we agree that we may have overstated "dynamic" in the previous manuscript, and so we have revised the statement in the revised manuscript

The work by Klie and Browning on (La,Sr)FeO_{3-x} mentioned by the referee reported that the cubic perovskite (La,Sr)FeO_{3-x} transforms into a brownmillerite phase with ordered oxygen vacancies by in situ heating under the reducing conditions

of the TEM column. In that study, microdomains grown with ordered vacant oxygen lattice sites and stacking faults of double FeO_2 layers were observed at a temperature of $800\text{ }^\circ\text{C}$. The high temperature structure of $(\text{La,Sr})\text{FeO}_{3-x}$ should be the brownmillerite phase with bright and dark columns. That is similar to the reversible topotactic phase changes between $\text{SrCoO}_{2.5}$ and $\text{SrCoO}_{3-\delta}$ depending on the amount of oxygen vacancies as shown in Ref. 10, where $\text{SrCoO}_{2.5}$ stabilizes in the brownmillerite phase with ordered oxygen vacancies while SrCoO_3 is in the cubic perovskite phase. However, this is not the situation in our work. In our work, the pristine state is the $\text{SrCoO}_{2.5}$ film with a brownmillerite structure, and we focused on the migration process of the oxygen vacancies and the subsequent influence on the structure evolution of $\text{SrCoO}_{2.5-x}$ when the oxygen vacancies were further extracted from the brownmillerite structure.

To further clarify, we performed an in situ heating experiment using the same sample configuration, and observed only a reorientation of the brownmillerite SCO film without any indication of an NSCO phase transformation under an electric field. As shown in the selected-area electron diffraction (SAED) pattern (Figure R4(a)), at room temperature, the diffraction spots $(0\ 0\ 1/2)_{\text{STO}}$ of the SCO film were observed and are indicated by yellow arrows. This was accompanied by the doubling of the cubic unit cell along the $[001]_{\text{STO}}$ direction. The repeated unit of the SCO film was outlined by a yellow rectangle with a short side in the $[001]_{\text{STO}}$ direction in Figure R4(a). The high-resolution transmission electron microscopy (HRTEM) image in Figure R4(b) also shows the doubled lattice planes in the $[001]_{\text{STO}}$ direction, as indicated by the two blue lines. However, when the temperature reached $400\text{ }^\circ\text{C}$ and higher, the diffraction spot $(0\ 0\ 1/2)_{\text{STO}}$ disappeared while the diffraction spot $(0\ 1/2\ 0)_{\text{STO}}$ appeared (Figure R4(c)). Accordingly, the doubled lattice planes in the $[010]_{\text{STO}}$ direction can be seen in the HRTEM image that is shown in Figure R4(d). In addition, the splitting of some of the diffraction spots along the $[001]_{\text{STO}}$ direction was observed in Figure R4(c), and this corresponds to the shorter in-plane lattice parameters of SCO relative to the STO substrate. When we increased the temperature to $900\text{ }^\circ\text{C}$, the film began to evaporate. Since the NSCO phase is characterized by the absence of the doubled unit cell and by the obvious elongation of the c axis, we believe that NSCO phase transformation certainly did not occur when there was only heat.

Figure R4 (unpublished). In-situ heating of the SCO film. SAED pattern (a) and HREM image (b) at room temperature; SAED pattern (c) and HREM image (d) at 400 degree.

Comment 1. In line 57, the authors seem to state that NCSI is a STEM based technique. This is clearly false, since negative CS imaging is only applicable for aberration-corrected TEM imaging.

Response 1: We thank the referee for pointing this out. The NCSI imaging was certainly performed in the TEM mode. We changed the wording “*scanning transmission electron microscopy (STEM)*” in lines 54-55 to “*transmission electron microscopy (TEM)*”.

Comment 2. The DFT modeling does not appear to reveal any information that was not previously known. Simply calculating the oxygen vacancy formation energies has been done numerous times in the reported literature.

Response 2: We agree with the referee that similar calculations of the oxygen vacancy formation energies have been reported in the literature. However, we

specifically performed the calculation in $\text{SrCoO}_{2.5}$ to distinguish theoretically which oxygen sites migrate more easily. Although this type of calculation is not new, it explains that the oxygen ions in the SrO layers have a higher tendency to be extracted in the brownmillerite $\text{SrCoO}_{2.5}$. To our knowledge, this has not been previously done in the $\text{SrCoO}_{2.5}$ system.

Comment 3. In Figure S2b), I would argue that the fit between the experiment and the modeled ABF images is poor. The contrast in the oxygen columns at 0.5, 1.4 and 2.1 nm is not reproduced at all in the models. If this serves as the foundation for the entire paper, then the authors should revise this model completely.

Response 3: We agree with the referee that there are some inconsistencies between the ABF images and the simulations. The contrasts in the oxygen columns at 0.5, 1.4, and 2.1 nm correspond to the Co sites in the tetrahedral layers. The contrast of Co varied at different sites. As shown in Fig. R5, the contrast of Co in the tetrahedral layers (Co_T) is relatively lower than that in the octahedral layers (Co_O). The contrast of Co also varied at different sites in the simulated images as shown in Figure R5. However, in the experimental images, factors such as sample thickness, local defects, and noise further affect the intensity of the atom column even at the same site. Thus, we consider that the mismatch between the ABF images and the simulated images is a result of those influences. Nevertheless, this mismatch does not change the basis and the conclusions of this paper. However, based on the referee's comment, we recalculated the ABF images with detailed simulations and obtained a better match between the ABF images and the simulations, and this is shown in Fig. R1. (We have also pasted it here).

Fig. R5. Structure model (a) and corresponding simulated ABF images with inversed line profile of contrast.

We compared the ABF images of simulations with oxygen occupancies of 0.4, 0.5, and 0.6, and found that 0.5 and 0.6 matched the decreased contrast in sites as

indicated by the green and blue arrows, respectively. Considering the fluctuations in the image contrast between the 0.5 and 0.6 cases, an error bar of 0.1 was given. Thus, y was estimated as 0.55 ± 0.1 . All of the results with error estimations can be found in the revised *Supplementary Table S1*.

Fig. R1. Comparison of ADF images of the 0V-LRS with the simulations on the oxygen occupancy of 0.4, 0.5 and 0.6 in the every other SrO planes.

Comment 4. How do the authors distinguish the reported new LCO phase from one that simply shows disordered oxygen vacancies? In a recent paper on PrYCaCoO_3 (Gulec, ACS Nano, 2016, 10(1): p. 938-947), the authors report a temperature induced oxygen vacancy ordering/disordering transition which is measured as a change in the sample conductivity. Is it possible that a similar transition is observed here rather than the formation of a new phase?

Response 4: We read the paper on PrYCaCoO_3 carefully and found the following differences:

- a. For PrYCaCoO_3 , “the atomic spacing clearly **decreases** at 90 K, by around 1.8%” while in our case the atomic spacing **increased** from $8.02 \pm 0.09 \text{ \AA}$ in the 1.0V-LRS state to $8.6 \pm 0.04 \text{ \AA}$ in the new NSCO phase as shown in Fig. 2h. Another distinct feature of the new NSCO phase was the large c/a ratio and the disappearance of the pre-peaks in the O K-edge, while the pre-peaks in PrYCaCoO_3 had only a slight shift.
- b. The O vacancy order/disorder transition observed in PrYCaCoO_3 was driven by the Co valence, **induced by the charge transfer in the Pr^{3+} - Pr^{4+} valence shift**, while the phase transition in our work was induced by **the change in the O content under an electric field**.

Reviewer #4 (Remarks to the Author):

Remarks to the Author. In situ observation of oxygen migration at the atomic scale is a really interesting topic that could answer many questions on the characterization of a variety of materials. **Such a novel approach could also provide important information on understanding the properties of systems as catalysts, membranes and fuel cells.** The authors claim to detect oxygen migration in brownmillerite-type SrCoO_{2.5-x} under electric field however I consider that there are some issues that need further explanation before considering the publication of the paper as a nature communication.

Comment 1. 1.- Page 6, (line 96 to 108) The authors claim to determine the oxygen occupancy in CoO_x and SrO_y layers by using the linear relationship reported in ref 12. It is not clear if they had used the linear relationship published in ref12, which covers a 3.5-4.8nm range or they have provided an experimental linear relationship for their own data (it could be included as supporting information). I think that this information needs to be clarified.

Response 1: We thank the referee for pointing this out. We provided an experimental linear relationship for our own data.

Here we used pristine SrCoO_{2.5} (-SrO-CoO1-SrO-CoO2-) as the reference to extract the linear coefficient. For x=1, d_{SrT}=4.44 Å, and for x=2, d_{SrO}=3.48 Å. We assumed a linear relationship between the oxygen content (x) and the lattice spacing (d) and thus obtained:

$$x=5.625-1.0416*d$$

From this, we can find the value of x in CoO_x by measuring the inter-planar spacings d_{SrT} and d_{SrO}. The results of the averaged occupations x in CoO_x layers for the LRS and 1V-LRS samples are shown in *Supplementary Table S1*. We added this information in the revised supporting materials.

Comment 2. 2.- Page 6 (line 110-117)

By comparing the ABF intensities on the pristine, LRS and 1V-LRS images, the experimental images show a change on the intensity periodicity between the pristine sample and the LRS and 1V-LRS samples. The authors claim that such a change indicates that **oxygen atoms were only extracted from every second Sr-O layer** on the LRS sample and a further reduction on the oxygen content is observed when applying 1V.

In order to justify such a result, the authors present ABF image simulations as supporting information. I think that this point needs a further explanation and some points need to be considered before arriving to such a conclusion.

Comment 2a. * They are comparing the experimental data Figure 2 i-k with the simulated image information (figure S2a, intensity profiles), however the experimental figure is displayed in nm and the simulated figure is displayed in pixels.

Response 2a: We thank the referee for pointing this out. The relationship between pixel and nanometer in the simulated figure is 120 pixel/nm. Accordingly, we revised the unit in the simulated figure to nm in the revision of Fig. S2b.

Comment 2b.* On the simulated data a sample thickness of 39nm is used. Why are they using such a value? Have they performed simulations at different thickness?

Response 2b: The value of 39 nm corresponds to a thickness of ~100 (pseudo-cubic) unit cells. In the simulation software we used (xHREM released by HREM Research Inc.), the thickness is scaled by slices. In our simulations we divided a cell into 1 slice, corresponding to 0.39 nm. Thus, we set the steps at thicknesses of 50, 100, 150, 200, and so on.

We performed simulations on the pristine SrCoO_{2.5} phase at different thicknesses as shown in Fig. R6. By comprehensively comparing the relative contrast of the Co and O columns, we found that a thickness of 100 slices was the most appropriate thickness. In addition, the sample milled using a focused-ion beam was around 30-50 nm, which was also consistent with using a thickness of 39 nm in the simulations. In the revised process, we did an additional series of simulations (with 90, 100, 110, and 120 slices) to confirm that 100 slices was the most appropriate value for the simulations.

Figure R6. Simulated ABF images with different thickness. Simulation condition: thickness: 39nm; collection angle for the ABF image: 11-22mrad. Probe size: 0.78nm. Defocus: 30 angstroms.

Comment 2c.* When comparing the pristine sample with the LRS and 1V-LRS samples, on the experimental data they mentioned a change on the intensity periodicity however if we look at the simulated intensity profiles (Fig S2); **such a change is not clearly observed from the pristine sample and the different oxygen occupancy images (0.1-0.9), at least, we need to go lower than 0.5 in order to see such a change in periodicity.** I think that a graph comparing all the experimental intensity profiles (pristine, LRS and 1-LRS) with the simulated one will help to understand such a critical point on the paper.

Response 2c: We thank the referee for pointing this out. We replotted the simulated images in two columns as shown in Fig. R7. Red arrows with different lengths were used to indicate the gradually decreasing intensities of the oxygen column in every other SrO layer. This indicates that oxygen atoms were only extracted from every other SrO layer. We also provided a graphical comparison of all of the experimental intensity profiles (pristine, LRS, and 1-LRS) below in the supplementary material.

Figure R7. Re-plotted simulated ABF images corresponding to different occupations of O_A at Sr_2O layers with a thickness of 39nm. The occupation of oxygen in every second SrO layer varied from 0.1 to 1. Simulation condition: thickness: 39nm; collection angle for the ABF image: 11-22mrad. Probe size: 0.78nm. Defocus: 30 angstrom.

Comment 2d. Moreover they indicated that the LRS state will correspond with $x=0.5$ occupancy (Figure S2 b). When looking at the Figure S2b, we observe that there is a discrepancy between the experimental and simulated every two layers (**between 0.5-0.75nm, between 1.25 and 1.5 nm etc...**). On the simulated images it is clear that the intensity corresponding to the first oxygen peak is disappearing when moving from the pristine to the 0.1 occupancy. **Why are they indicating that LRS sample corresponds to an occupancy of $x=0.5$ and not for example to 0.6 or 0.7?** I think that it is important to clarify this part of the paper.

Response 2d: We agree with the referee that there are discrepancies between the experimental and simulated results (between 0.5-0.75 nm, and between 1.25 and 1.5nm), and this corresponds to the O columns with an occupancy of 1. The absence of peaks corresponding to the oxygen columns in the two positions between 0.5-0.75 nm, and between 1.25 and 1.5 nm should correspond to the relatively diffuse and broad ABF contrast of the neighboring Co columns, which blurred the contrast of the O columns. If there is large deviation in the full occupancy between 2.0-2.25 nm and 2.75-3.0 nm, there should be an obvious valley in the neighboring O columns.

In the revised manuscript, we estimated an approximate range for the oxygen occupancy in the SrO planes via a systematic comparison. For example, as shown in Fig. R1, we compared the ABF images to the simulations with oxygen occupancies of 0.4, 0.5, and 0.6 and found that the simulations with 0.5 and 0.6 matched the decreased contrast in sites as indicated by the green and blue arrows, respectively. Considering the fluctuations in the image contrast between the 0.5 and 0.6 cases, an error bar of 0.1 was given. Thus, y was estimated as 0.55 ± 0.1 . All of the results with error estimates can be found in the revised *Supplementary Table S1*.

Fig. R8. Comparison of ABF images of the pristine, 0V-LRS and 1V-LRS with the simulations on the different oxygen occupancy in the every other SrO planes.

Comment 2e.* Do they have any analytical information (as for example EELS quantification) to support the different oxygen content between the pristine and LRS, 1V-LRS samples?

Response 2e: In the revised version, we have provided detailed EELS quantification of the Co valence state, and this is discussed in the following Response 3.

Comment 3a. 3.- Page 8 Nano-twinned NSCO phase. When applying a 1.5V the authors reported the formation of a new crystal structure.

On Figure 3, they present EELS data from the pristine, LRS and NSCO samples. I understand that such information has also been extracted from the 1V-LRS sample.

How much is the oxygen content on the indicated samples? What are the experimental conditions for the EELS acquisition?

Response 3a:

EELS spectrum of LRS:

The EELS spectrum of LRS is shown in Fig. 3d and was acquired using the 0V-LRS sample. The oxygen content was about 2.26 ± 0.14 , and this was measured from the HAADF/ABF images. The value of $2.54 \sim 2.67$ was determined by analyzing the Co $L_{2,3}$ ratio. Unfortunately, we did not record the EELS in the 1V state since switching between the image mode and the EELS mode requires changing the beam current setting. Such a change takes several minutes, and the details of the structural evolutions would be missed. Instead, we acquired the HAADF/ABF images, and from these, the oxygen content was determined to be 2.09 ± 0.16 .

EELS experimental conditions:

The EELS experiments were carried out with a Gatan spectrometer attached to the TEM in the STEM mode operating at 200 kV. The convergence semi-angle for the electron probe was 22 mrad. The spectrometer was set to an energy dispersion of 0.25 eV/channel. All of the spectra were acquired with a short exposure time, which was as short as 0.05 s, in a line-scan manner. Three hundred spectra were summed to avoid electron-beam damage.

Comment 3b. On line 153, “The first peak (529.5 eV) at the O K edge decreased when the sample changed from its pristine state to LRS, indicating the creation of V_{O_S} [23]” In ref 23, Gazquez et al, after analyzing the $L_{3,2}$ ratio, they conclude that the oxidation state is close to Co^{+2} and it remains unchanged, moreover they conclude that the measured spatial modulation in the O-K edge prepeak is the fingerprint for a spin-state superlattice caused by the ordering of high concentration of O vacancies. On the present paper (line 153) the authors claim that the first peak at the O K edge decreased when the sample change from its pristine state to LRS. Such an observation is not clearly observed on Figure 3d. The figure needs to be modified in order to better appreciate such a change on the figure.

Response 3b: According to the referee’s suggestion, we have modified Fig. 3d by plotting three groups of spectra without an offset to enable comparing the pre-peak of the O K edge and the Co L_2 edge with normalized intensities. A decrease in the pre-peak of the O K edge was clearly observed, and this corresponded to a change in the material from its pristine state to LRS.

Ref. 23 was indeed not appropriately cited here. We have removed this reference and cited the earlier work by Prof. Klie and Prof. Browning on $SrCoO_{3-\delta}$ (Ultramicroscopy 86 (2001) 289–302), where the pre-peak of the O K edge and the L_3/L_2 intensity ratio of Co were discussed.

Comment 3c. Moreover on figure 3d, they indicated that the L_3 peak of Co goes from 772.5eV on the NSCO sample to 774.50 on LRS sample and 777.20eV on

pristine sample indicating that the valence decrease of Co+3 on NSCO sample. In this context I have to mentioned that there are several papers that shown that a change in valence state of cations introduces a dramatic change in the $L_{2,3}$ white line ratio (see for example Pearson et al. Appl. Phys. Lett. 53, 1405–1407, 1998, Pearson et al. Phys. Rev. B 47, 8471–8478. 1993; Kurata and Colliex Phys. Rev B, 48, 2102-2108) In particular Z.L. Wang et al. Micron 31 (2000) 571–580) 2000, reported the behavior of several cobalt oxides with oxidation states going from +2 to +4 and they also determine the oxidation state of a perovskite-type crystals.

Response 3c: According to the referee’s suggestion, we analyzed the $L_{2,3}$ ratio with the method reported by Prof. Z.L. Wang et al. [Micron 31 (2000) 571–580]. An empirical fitting curve of the Co valence state as a function of the Co L_3/L_2 ratio was plotted and is shown below. Ratio values of 2.82 ± 0.15 , 3.32 ± 0.11 , and 4.61 ± 0.23 were obtained for the pristine, 0V-LRS, and NSCO phases, respectively, based on the statistical analysis of more than ten spectra of each sample. As shown in Fig. R9, we plotted our data on the empirical fitting curve to estimate the valence state of Co in the three states, and the values of 2.84~3.08, 2.54~2.67, and 2.02~2.15 corresponded to the pristine, 0V-LRS, and NSCO phases, respectively.

Figure R9. Comparative plot to estimate the valence state of Co in the pristine, 0V-LRS state and the NSCO phase (blue circles with error bars). The dotted line is the empirical fitting curve of Co valence state as a function of Co L_3/L_2 ratio with reference to the known standard samples (solid triangles) established by Wang et al.

We also estimated the oxygen content from the valence states of Co using the electroneutrality principle. We obtained oxygen contents of 2.42~2.54, 2.27~2.34, and 2.01~2.07 for the pristine, 0V-LRS, and NSCO phases, respectively. These are in good agreement with the results from the analysis of the HAADF and ABF images,

which are 2.50 ± 0.10 , 2.26 ± 0.14 , and 2.09 ± 0.16 for the pristine, 0V-LRS, and 1V-LRS states, respectively.

Comment 3d. On the mentioned papers and at the EELS spectra published on the EELS database <https://eelsdb.eu/> we can see that the L_{2,3} signal of several Co compounds are on the 780-790eV range. I suggest to the authors to analyze the L_{2,3} ratio in order to determine the oxidation state on their samples and to check the energy value of the Co L_{2,3} signal on the Figure 3d.

When comparing the Co L_{2,3} on the pristine and LRS samples there is a shift in energy from 777.20eV towards 774.50. What does mean? How they can explain such a modification?

Response 3d: We compared the peak position of Co L₃ with several reports in the literature (for example, PRL 99, 047203 (2007) and Chem. Mater. 26, 2496 (2014)) and found that the pre-peak of the O K edge was around 531~533 eV, and Co L₃ was at 780~782 eV. In our spectra, the pre-peak of the O K edge was at 529.5 eV, and Co L₃ was at 777.2 eV. In fact, we calibrated the pre-peak of the O K edge using the Ti L edges of the SrTiO₃ substrate, which we acquired simultaneously in the same line-scan spectra.

In addition, there are also some reports in the literature in which the value of the Co L₃ signal is below 780 eV. For example, in a recent paper on (Pr_{0.85}Y_{0.15})_{0.7}Ca_{0.3}CoO_{3-δ} (Gulec, ACS Nano, 2016. 10(1): p. 938-947), the energy value of the O pre-peak and the Co L₃ signal were ~533 eV and ~776.5 eV, respectively, as shown below.

Figure. R10. The Fig. 2a and 2b in the (Pr_{0.85}Y_{0.15})_{0.7}Ca_{0.3}CoO_{3-δ} paper (Gulec, ACS Nano, 2016. 10(1): p. 938-947).

We also checked the energy value of the Co L₃ signal from the EELS database (<https://eelsdb.eu/>) and found the following:

792.5 eV in LaCoO₃ submitted by Ferdinand Hofer, January 19, 2008.

782.8 eV in SrCoO Misfit NT submitted by Luc Lajaunie, December 19, 2016.

781.0 eV in CoO submitted by Yuan Zhao, July 20, 2014.

787.3 eV in Co₃O₄ submitted by Xu Qiang, December 11, 2007.

We found that the energy value of the Co L_3 signal varied widely with different Co compounds despite a smaller change in the valence state of Co, and this demonstrates that the energy value of the Co L_3 signal is very sensitive to its chemical environment. In our work, the brownmillerite $\text{SrCoO}_{2.5}$ had one ordered oxygen vacancy in each tetrahedral layer. From the pristine state to LRS, the oxygen in every other SrO plane was extracted, and this significantly changed the coordination environment of Co. This might be the reason for the obvious energy shift from the pristine state to LRS. The similar situation is expected in the shift from LRS to NSCO, where an obvious lattice change should also affect the chemical environment of Co. Thus, we decided to use the L_3/L_2 ratio, which is more robust, to determine the valence state of the Co ions.

Comment 4. 4.- I also consider that some modifications could be included in the figures in order to clarify the diagram. In figure 1 c, it will be interesting to indicate at the figure where the sample $\text{SrCoO}_{2.5}$ is located at the experimental set-up.

Response 4: We thank the referee for this good suggestion. In order to clarify the diagram, we have revised Fig. 1b. Specifically, we have added a low magnification image showing the film and substrate, and thus bridge the gap between the SEM microscope and the atomic-resolution HAADF image.

Figure R10. Revised figures for the in-situ experimental setup, where the sample position is indicated by the “Film”.

REVIEWERS' COMMENTS:

Reviewer #1 (Remarks to the Author):

The authors have addressed my comments and suggestions.

Reviewer #2 (Remarks to the Author):

All my concerns have been addressed. On a final note, the authors should consider rewording the second sentence of the abstract. It is not clear why batteries are mentioned here and why visualizing oxygen vacancy migration at atomic resolution is important for any battery research.

Reviewer #4 (Remarks to the Author):

I would like to point out the work done by the authors in order to answer the different comments included by the referee.

I consider that the authors have provided additional information that answers the different remarks. However, I would like to mention a few points that need to be corrected or clarified:

1.- In page 6 lines 103-104:

"distances, d_{SrT} and d_{SrO} (indicated by three horizontal yellow lines as shown on HAADF images in Fig. 2a-c), which correspond to the spacings of octahedral and tetrahedral layers, respectively."

I understand that the proper sentence will be...

which correspond to the spacings of tetrahedral and octahedral layers, respectively

2.- The authors have modified figure 3d.

At the present figure, the Co peaks are at the same position in energy, however, in the text (line 160-162) they indicated

"Meanwhile, the Co L3 edge shifts towards a lower energy of 161 774.5 eV and 772.7 eV in the LRS and NSCO (see Supplementary Fig. S8), respectively, 162 indicating an obvious change in the electronic structure of the NSCO phase."

Moreover, in the figure caption they claim "The peak positions of 169 Co L3 have been marked in each spectrum"

From the supporting information I understand that there is a clear change in energy on the Co L3 edge. I consider that the present figure displays much better the O-K pre-peak variations; however, they must explain or indicate (in the text and figure caption) that they have modified the figure energy position to better display the intensity L3, L2 variations @ Co L2,3 edge

Reviewer #2 (Remarks to the Author):

All my concerns have been addressed. On a final note, the authors should consider rewording the second sentence of the abstract. It is not clear why batteries are mentioned here and why visualizing oxygen vacancy migration at atomic resolution is important for any battery research.

Response:

According to reviewer's suggestion, we have deleted the statement about "batteres" in the second sentence.

Reviewer #4 (Remarks to the Author):

I would like to point out the work done by the authors in order to answer the different comments included by the referee.

I consider that the authors have provided additional information that answer the different remarks. However I would like to mention to points that need to be corrected or clarified:

Comment 1.-In page 6 lines 103-104:

"distances, dSrT and dSrO (indicated by three horizontal yellow lines as shown on HAADF images in Fig. 2a-c), which correspond to the spacings of octahedral and tetrahedral layers, respectively."

I understand that the proper sentence will be...

which correspond to the spacings of tetrahedral and octahedral layers, respectively

Response 1:

We thank the referee for pointing this out! We have changed the sentence "which correspond to the spacings of octahedral and tetrahedral layers, respectively" to "which correspond to the spacings of tetrahedral and octahedral layers, respectively".

Comment 2.- The authors have modified figure 3d.

At the present figure, the Co peaks are at the same position in energy, however on the text (line 160-162) they indicated "Meanwhile, the Co L3 edge shifts towards a lower energy of 774.5 eV and 772.7 eV in the LRS and NSCO (see Supplementary Fig. S8), respectively, indicating an obvious change in the electronic structure of the NSCO phase." Moreover at the figure caption they claim "The peak positions of Co L3 have been marked in each spectrum" From the supporting information I understand that there is a clear change in energy on the Co L3 edge. I consider that the present figure display much better the O-K pre-peak variations however they must explain or indicate (on the text and caption figure) that they have modified the figure energy position to better display the Intensity L3, L2 variations @ Co L2,3 edge

Response 2:

We have changed the figure caption to "The peak positions of Co L₃ edge in each spectrum have been shifted to the same energy position in order to better display the intensity variations of Co L₃ edge.". We also explained our modification on the energy position in the text by adding the sentence "Here we aligned the peak positions of Co L₃ edge in each spectrum to

better display the intensity variations of Co L₂ edge.”.